computational biology/bioengineering/
biomechanics

touch, biomechanics, neuroscience,
computational modelling

**Author for correspondence:**
Victor H. Barocas
e-mail: baroc001@umn.edu

# An inter-species computational analysis of vibrotactile sensitivity in Pacinian and Herbst corpuscles

Julia C. Quindlen-Hotek, Ellen T. Bloom,
Olivia K. Johnston and Victor H. Barocas

Department of Biomedical Engineering, University of Minnesota, Minneapolis, MN, USA

 JCQ-H, 0000-0001-9198-3011; ETB, 0000-0003-4882-9793;
VHB, 0000-0003-4964-2533

Vibration sensing is ubiquitous among vertebrates, with the sensory end organ generally being a multilayered ellipsoidal structure. There is, however, a wide range of sizes and structural arrangements across species. In this work, we applied our earlier computational model of the Pacinian corpuscle to predict the sensory response of different species to various stimulus frequencies, and based on the results, we identified the optimal frequency for vibration sensing and the bandwidth over which frequencies should be most detectable. We found that although the size and layering of the corpuscles were very different, almost all of the 19 species studied showed very similar sensitivity ranges. The human and goose were the notable exceptions, with their corpuscle tuned to higher frequencies (130–170 versus 40–50 Hz). We observed no correlation between animal size and any measure of corpuscle geometry in our model. Based on the results generated by our computational model, we hypothesize that lamellar corpuscles across different species may use different sizes and structures to achieve similar frequency detection bands.

## 1. Introduction

Vibrotactile sensitivity across different animal types is fine-tuned to fit each species' particular needs [1]. Species rely on vibrotactile sensitivity for various purposes, ranging from hunting to predator detection to tool manipulation. The Pacinian corpuscle (PC) is a cutaneous mechanoreceptor responsible for sensing

high frequency (20–1000 Hz) vibrations [2–4]. The PC has been reported in species such as mammals, reptiles and amphibians and has been identified in many anatomical locations, including human hands [5] and elephant feet [6]. The PC's avian homologue, the Herbst corpuscle (HC), is found primarily in the bird's beak but also in other regions such as the legs [7,8]. These homologous end organs show functional similarities but some structural differences, such as inner core organization [4,7]. However, both the PC and HC are ovoid receptors comprising a single myelinated nerve fibre encapsulated by collagenous lamellae [7]. In this paper, we use the term 'lamellar corpuscles' to refer collectively to Pacinian and Herbst corpuscles.

Specialization in the structural properties of the lamellar corpuscles, as well as their anatomical location, enables these end organs to serve distinct sensory roles for different species. The somatosensory organs of snakes are specialized to aid in hunting, as the snakes must rely on their sense of touch to hunt and catch fish in complete darkness and may use vibrations of surrounding water to locate prey [1,9]. Crocodiles also rely on their sense of touch to hunt in dark environments [1]. The star-nosed mole uses its star organ, which includes 22 mechanosensitive appendages around the nostrils, to forage in muddy soil [10]. Each appendage is innervated by various mechanoreceptors, including Pacinian-like corpuscles, and contains the densest population of mechanosensitive end organs found in mammals [1,10,11].

Birds rely on tactile sensitivity within their skin and beaks or bills. Foragers, such as kiwi and sandpipers, rely on mechanosensors in their bills to locate invertebrates in sand and soil via 'remote touch' to sense vibrations generated by burrowing prey [1,12]. Anseriformes, including ducks, geese and swans, use their touch to hunt in aquatic environments. Some species of ducks use 'dabbling,' during which they move their mechanosensitive bill back and forth in water, to forage [1].

Mammalian PCs are also implicated in other functions depending on anatomical location. The PCs of human and monkey hands are localized to areas necessary for gripping and tool manipulation [13]. Elephants are believed to use PCs in their feet to sense low-frequency seismic waves for long-distance communication [6], and kangaroos may use PCs located in their extremities to sense ground vibrations, which could aid in predator detection [14].

Given the wide range of lamellar corpuscle size, structure and function across species, one must ask how the size and structure vary, and what the functional consequences of this variation are. Thus, the goal of this study was to use our previously published theoretical model of the PC [15–17] to predict functional differences among lamellar corpuscles of different species based on structural differences. A literature search was performed to obtain lamellar corpuscle outer core structural parameters. Next, a three-stage model of the PC's mechano-to-neural transduction was used in a species-specific manner to simulate neural responses resulting from surface vibrations. Tuning curves were generated for each animal, and the peak frequency and bandwidth of these tuning curves were compared across species.

# 2. Material and methods

## 2.1. Modelling scheme

A previously published [15] computational model of the PC's mechano-to-neural transduction was used to simulate the PC's neural response to vibration. This model comprises three stages, run in series, each dealing with specific components of the PC's transduction process. The input to Stage 1 is a mechanical vibration applied to the PC's outer surface and the output of Stage 3 is the spike train elicited by that vibratory stimulus, as might be recorded *in vivo*.

Stage 1 of the model is a finite-element mechanical model of the PC's outer core. The outer core is modelled as a sphere with alternating layers of fluid and solid to represent the inner structure. The mechanics of the lamellae and inter-lamellar fluid are based on shell and lubrication theory, respectively. The lamellae are assigned a Young's modulus, thickness and radial location, and the fluid layers are assigned viscosity and thickness. The outermost shell of the Stage 1 model is vibrated with a spatiotemporal pressure wave whose peaks occur at the poles and equator of the corpuscle and whose temporal frequency is specified (see equation 31 of [15] for more detail). The resulting deformation of the innermost shell is obtained and used as the boundary condition for Stage 2.

Stage 2 is a second finite-element mechanical model of the PC's inner core and neurite in COMSOL. Five filopodia protrude from the neurite's surface. These filopodial bases are hypothesized to be the locations of mechanically gated cation channels critical to the PC's vibration response [18]. The outer surface of the inner core is deformed according to the Stage 1 output, and the resulting neurite surface strains are passed through a saturation function [19] to convert strain to channel current. This

current is then input to Stage 3, which is a computational electrochemical neurite model in NEURON [20]. The Stage 3 model simulates the opening of mechanically gated cation channels on the neurite's membrane. The resulting action potentials are recorded, and the firing rates generated from vibratory inputs at various amplitudes are calculated. Combining the three stages, the computational model allows simulation of the PC's neural response to an arbitrary vibratory mechanical stimulus applied to the receptor's outer surface

## 2.2. Structural analysis of corpuscles

A literature search was performed to obtain lamellar corpuscle structural parameters. Species were selected based on the availability of relevant data and quality of published histological images. For the cat PC, the number of outer core lamellae, lamellar thickness and corpuscle outer radius were obtained from published findings [21,22]. For the duck HC, the number of lamellae was obtained from published text [23,24]. In most cases, a diameter range was not reported by the paper in question, so all other species' structural properties were measured by analysis of histological images using ImageJ [25,26]. In some cases, only one histological image of the corpuscle was provided; in cases where the paper in question reported multiple images, the image with the clearest lamellar organization was chosen for analysis. Outer core lamellae were counted manually; corpuscle outer radius and lamellar thickness were measured based on image scale bars. If no scale bar was provided, corpuscle radius and lamellar thickness were measured in pixels, since the computational model can be run using only the ratio between thickness and radius when actual values are unavailable. The lamellar thicknesses estimated in this approach depended on the resolution of the provided image and did not include factors such as the connective tissues, collagen fibrils and proteoglycans present between lamellae [27,28].

Linear regressions were performed to compare the statistical relationships between animal size and corpuscle structural parameters. The $p$-values were calculated to test the null hypothesis that animal size had no effect on corpuscle radius, lamellar thickness or number of lamellae. Linear regressions were also performed to test the relationship between corpuscle outer radius and the thickness and number of lamellae. The $p$-values were calculated to test the null hypothesis that corpuscle outer radius had no effect on lamellar thickness or number of lamellae.

## 2.3. Mechanical and neural simulations

The measured lamellar corpuscle radius, lamellar thickness and number of lamellae were used as structural inputs to Stage 1. The elastic (Young's) modulus for a lamella was set at 1.4 kPa, following our previous experiments on the human PC [29]. Inter-lamellar fluid viscosity was specified to be 3.5 mPa s, or approximately 5 times that of water at 37°C [30]. Stages 1–3 of the multiphysics PC model were run for stimuli at 2–1000 Hz.

Neural tuning curves were generated for each animal by determining the surface displacement at which the evoked neural firing frequency equalled the surface indentation frequency. We defined the *peak frequency* to be the minimum of the tuning curve, or the frequency at which the tuning threshold amplitude was minimized ($A_{min}$). Our other measure, the *bandwidth*, was defined as the frequency range over which the tuning threshold was less than $3.5A_{min}$. This cut-off was determined by comparing simulated human results with the reported functional human bandwidth of 40–1000 Hz [4,31]. When a lower bandwidth limit could not be measured because the tuning threshold remained low even at very low frequencies, the limit was set at 0 Hz.

# 3. Results

Usable images of lamellar corpuscles were found for 19 animals; structural data calculated from those images are summarized in table 1 along with the calculated peak frequency and bandwidth from neural tuning curves simulated for each animal. For the rat, we used images provided by B. Güçlü. Table 1 contains a single value, and not a range, for each parameter reported for different species. While anatomical variation exists between corpuscles in a single species and has been reported in the literature [21,27,28,39], these detailed evaluations of corpuscle structure and published ranges have mainly been reported for the cat corpuscle. Therefore, for animal species other than the cat and duck, a single histological image was analysed and the parameters determined from that one image are

Table 1. Species and corpuscle information. The data for the rat are based on images provided by B. Güçlü (Bogaziçi University).

| animal information | | Pacinian corpuscle properties | | | | computed features | | ref. |
|---|---|---|---|---|---|---|---|---|
| common name | species name | mass (kg) | outer radius (µm) | number of lamellae | average lamellar thickness (µm) | peak frequency (Hz) | bandwidth (Hz) | |
| cat | Felis catus | 4.04 | 255.6 | 30 | 0.2 | 48 | 142.9 | [21,22] |
| crocodile | Crocodylus niloticus | 386 | 75.14 | 10 | 3.18 | 43 | 168.4 | [44] |
| dog | Canis familiaris | 46.5 | 1971.24[c] | 10 | 10.92[c] | 21 | 174.7 | [32] |
| duck | Anas platyrhynchos | 1.15 | 43.9 | 15[a] | 0.91 | 41.5 | 154 | [23,24] |
| elephant | Elephas maximus | 4082 | 317.3 | 26 | 6.13 | 48 | 177.6 | [6] |
| emu | Dromaius novaehollandiae | 37.9 | 61 | 15 | 1.77 | 40 | 168.5 | [45] |
| frog | Rana esculata | 0.04 | 59.99 | 15 | 1.60 | 30 | 160.5 | [46] |
| goose | Anserini | 4.38 | 66.02 | 30 | 0.51 | 165 | 228 | [33] |
| human | Homo sapien | 62.1 | 190 | 28 | 1.1 | 137.5 | 683.7 | [15] |
| kangaroo | Macropus giganteus | 90.7 | 240.7 | 15 | 4.14 | 44 | 154.4 | [14] |
| mole | Scapanus orarius | 0.055 | 4.6[b] | 10 | 0.11 | 40.5 | 168.3 | [34] |
| monkey | Ateles fusciceps | 1.81 | 720.9[c] | 10 | 16.92[c] | 42 | 178.2 | [35] |
| mouse | Mus musculus | 0.0193 | 54.67 | 12 | 2.77 | 40 | 180 | [36] |
| ostrich | Struthio camelus | 107 | 37.51 | 7 | 3.20 | 20.5 | 170.3 | [45] |
| porpoise | Phocoena phocoena | 54.4 | 18.6 | 10 | 0.89 | 41.5 | 200 | [37] |
| rat | Rattus | 0.23 | 92.67 | 18 | 1.97 | 43 | 137.7 | |
| rooster | Gallus gallus | 3.45 | 203.8[c] | 8 | 4.57[c] | 42 | 177.3 | [47] |
| snake | Elaphe quadrivirgata | 0.33 | 977.6[c] | 18 | 5.30[c] | 45 | 146.4 | [48–50] |
| whale | Eschrichtius robustus | 36 000 | 74.02 | 9 | 3.10 | 42 | 188.5 | [38] |

[a]The number of lamellae was approximated as 15 based on text from two published studies [23,24].
[b]While the diameter of the lamellar corpuscle in the star-nosed mole was previously reported as 17.5 µm [34], no additional structural information was provided. Therefore, the diameter was measured from a micrograph of a PC included in that study, yielding a diameter of 9.2 µm.
[c]Measurements are in pixels because no scale was provided in the published images.

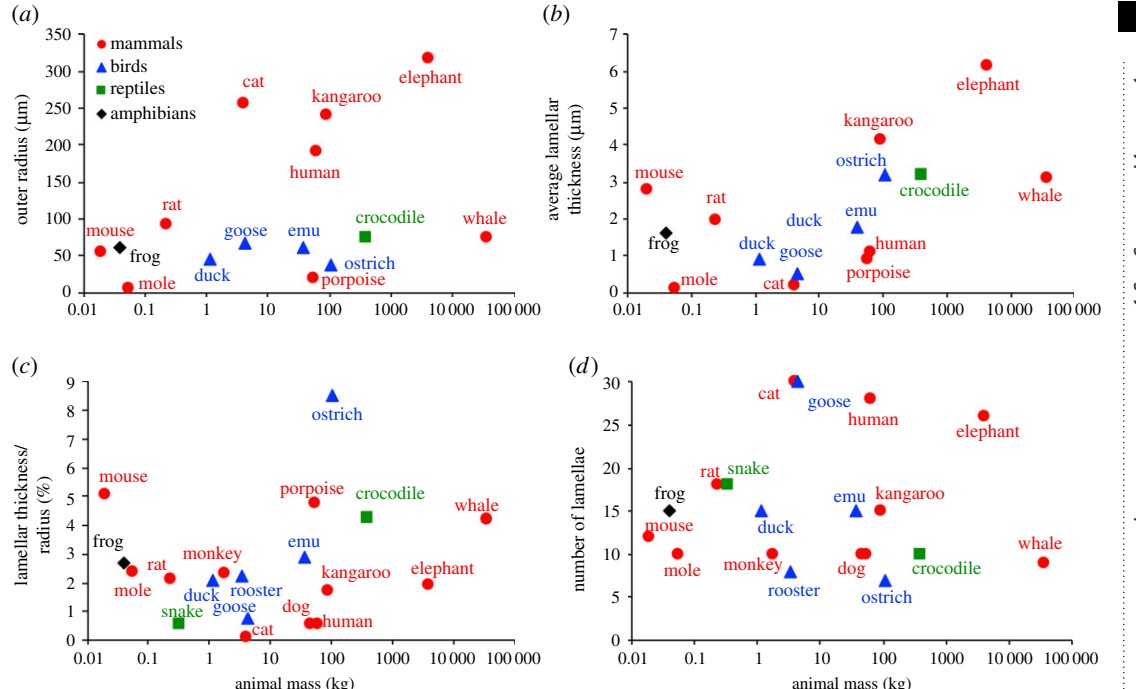

**Figure 1.** Lamellar corpuscle structural parameters. (*a*) Corpuscle outer radius versus animal mass. The colour and shape of the data points reflect taxonomic class. Outer radii listed in pixels in table 1 are excluded. (*b*) Average lamellar thickness in the outer core versus animal mass. Lamellar thicknesses listed in pixels in table 1 are excluded. (*c*) Average lamellar thickness/corpuscle radius versus animal mass for all animals, including those excluded from (*a,b*). (*d*) Number of outer core lamellae versus animal mass. None of the plots shows a consistent trend with animal mass, and the wide structural variability across species is evident.

reported in table 1. The data are shown graphically in figure 1 to visualize the relationship or lack thereof between animal size and corpuscle size. Corpuscle outer radius and average lamellar thickness for each animal are shown in figure 1*a,b* for the cases in which the histological image included a scale bar. Lamellar thickness was normalized by corpuscle outer radius in figure 1*c* for all animals. The number of lamellae counted in the outer shell is shown in figure 1*d*. There was no significant correlation with animal mass or taxonomic class for any calculated quantity ($p > 0.38$ for all cases).

The corpuscular structural properties listed in table 1 were analysed to determine whether a corpuscle's outer radius correlated with the thickness or number of lamellae in that corpuscle. In contrast with the lack of effect of animal mass on corpuscle structure, corpuscle outer radius was a good predictor of lamellar organization. Average lamellar thickness is moderately ($p = 0.07$; $r^2 = 0.23$) correlated with corpuscle outer radius (figure 2*a*), and the number of lamellae in a corpuscle is strongly ($p < 0.01$, $r^2 = 0.43$) correlated with the outer radius of that corpuscle (figure 2*b*).

This model has previously been compared with the functional response in the cat [15] and to psychophysical responses in human subjects [40] but has not been validated with an avian species. Simulations were performed using published micrographs of HCs in duck [23,24] and goose bills [33]. The simulated tuning curve of the duck corpuscle was compared (figure 3) with published electrophysiological tuning curves from seven HCs [41], with the caveat that the neurophysiological and structural studies were performed years apart and on different individual animals; unfortunately, the neurophysiological study [41] included no structural data, and the structural studies [23,24] included no neurophysiological data. The neurophysiological data [41] plotted in figure 3 were obtained from seven HCs that responded to 1–1075 Hz vibrations induced by a 0.5 mm diameter probe on the bill surface. Detailed methods for the experiment can be found in the published manuscript by Gregory [41]. The simulated tuning curve shows a lower peak frequency than the published tuning curves, but it falls comfortably within the range of observed thresholds despite using no adjustable parameters. Furthermore, the physical properties and *in vivo* depths for the HCs tested by Gregory were not reported [41], and knowledge of these parameters would provide insight into the experimentally obtained tuning curves and how they compare with the simulated response. The computational curve also has much steeper slopes at the boundaries of the receptive range than the experimental curves, possibly due to an effect from the surrounding tissue in the experiments that

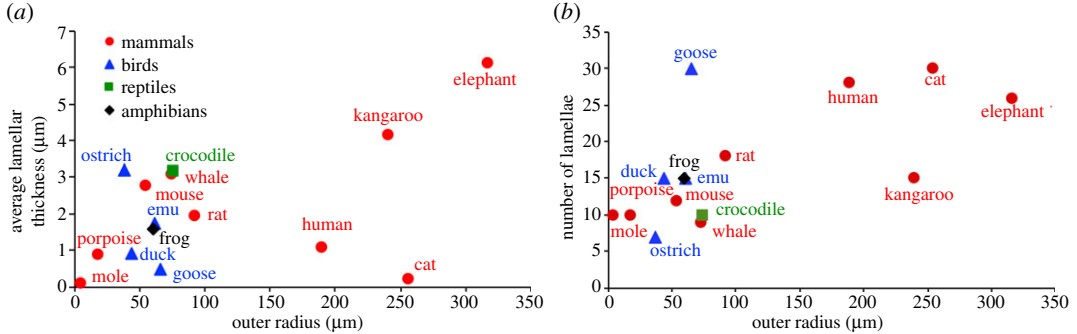

**Figure 2.** Corpuscle radius versus lamellar parameters. (*a*) Corpuscle outer radius versus average lamellar thickness. Lamellar thicknesses listed in pixels in table 1 are excluded. The lamellar thickness shows a slight ($p = 0.07$, $r^2 = 0.23$) correlation with corpuscle radius. (*b*) Corpuscle outer radius versus the number of outer core lamellae. There is a strong correlation ($p < 0.01$, $r^2 = 0.43$) between the number of lamellae and corpuscle radius.

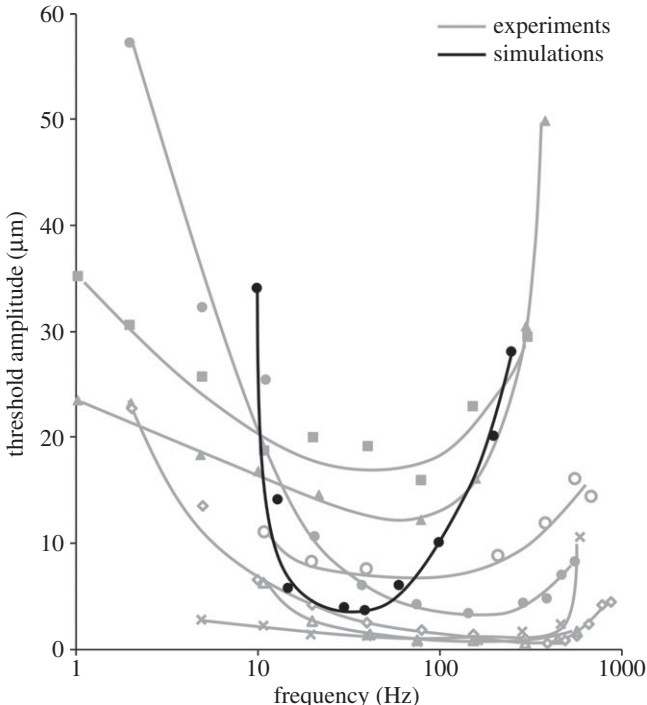

**Figure 3.** Comparison of simulated tuning curve for the duck Herbst corpuscle to published electrophysiological data [41] collected from Herbst corpuscles.

was not accounted for in the corpuscle-only simulations. In particular, the distance between the stimulus and the corpuscle could not be controlled in Gregory's experiments [41], leading to potential attenuation of signal [17,42]. Accounting for the surrounding tissue in our computational model would better match the simulated tuning curve for the duck HC to the published experimental results shown in figure 3. No tuning curves have been published for HCs in the goose bill, but it has been reported that goose-bill mechanoreceptors sensitive to sinusoidal stimuli exhibited peak sensitivity between 300 and 600 Hz [43]. The goose HC simulated in this study had a peak frequency of 165 Hz (figure 4*a*), which is below that reported peak sensitivity range [43]. The simulated goose HC response, however, is higher than that for all other simulated animals (figure 4*a*), which is consistent with the comparably high reported sensitivity range for the goose [43]. Furthermore, our simulations predict a higher peak frequency for the goose (165 Hz) compared with the cat (48 Hz), which is consistent with published reports that indicate that the goose corpuscle [43] is sensitive to higher frequencies than that of the cat [3]. These comparisons suggest that this model can provide a good prediction of the vibrotactile sensitivity range of an animal's lamellar corpuscle, in spite of various potential factors that we ignored in our calculations (e.g. differences in neuron electrophysiology).

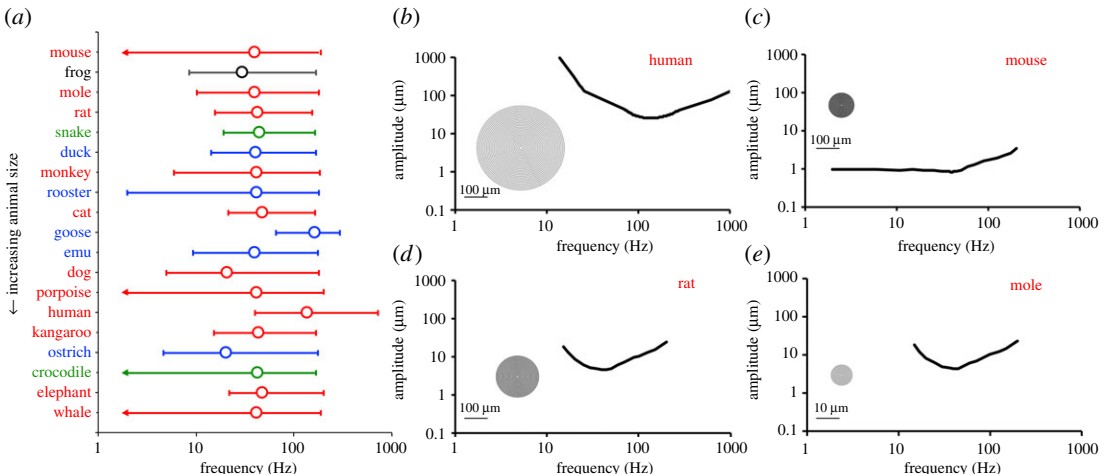

**Figure 4.** Simulated neural response of corpuscles. (*a*) Peak frequency, defined as the frequency at which amplitude is minimized ($A_{min}$), and bandwidth, defined as the frequency range over which amplitude is less than 3.5$A_{min}$. Peak frequency is labelled with an open circle. Upper and lower bandwidth limits are labelled by horizontal error bars with caps. In the cases where a lower bandwidth limit could not be identified, the lowest frequency tested that resulted in a tuning threshold less than 3.5$A_{min}$ is indicated by an arrow. For almost all animals, the peak frequency for corpuscle performance is in the 40–50 Hz range, with the human (high) and the dog and ostrich (low) being notable exceptions. (*b*–*e*) Tuning curves and representative PCs for (*b*) human, (*c*) mouse, (*d*) rat and (*e*) mole. Inserts show schematics of the PC structure based on properties in table 1. The scale bar for the mole (*e*) is 10× smaller than the other three.

The simulations were computed for the other species, for which we do not have published electrophysiological data. Peak frequencies and bandwidths for each animal are shown in figure 4*a*. Tuning curves for four selected animals (human, mouse, rat and mole) are shown in figure 4*b*–*e* with representative drawings of PC structure. In contrast with the highly variable structural data of figure 1, the tuning data of figure 4*a* are remarkably consistent. About 14 of the 19 species showed peak frequencies in the 40–50 Hz range in spite of structural differences. The inter-species consistency can be seen easily by comparing the tuning curves of the rat (figure 4*d*) and the mole (figure 4*e*). The rat's PC is 20 times larger than the mole's, has almost twice as many lamellae and has lamellae almost 20 times as thick. These differences balance in such a way as to produce almost identical tuning curves.

## 4. Discussion

Our key finding is that although the different species have a wide range of lamellar corpuscle size and structure, the simulated peak frequencies for nearly all of the corpuscles tend to cluster in the 20–50 Hz frequency range, with a preference for 40–50 Hz. This range was achieved in different animals by very different corpuscular structures. For example, the lamellar layer thickness and PC outer radius in kangaroo corpuscles are twice those found in the rat (figure 1*a*,*b*), but the two PCs produce similar frequency responses (figure 4*a*). While the rooster's mass is 400 times that of the rat and its lamellar corpuscle is also larger, the two animals possess similar lamellar counts and lamellar thickness to corpuscle radius ratios (figure 1*c*,*d*). The notable exceptions to this frequency focusing are the human, with a peak frequency of 135–140 Hz, and the goose, with a peak frequency of 165–170 Hz. The human hand serves many different sensory purposes that are distinct from those of other animals in table 1, which may require higher frequency sensitivity. The potential evolutionary advantages of the higher frequency tuning in the human PC merit future exploration. Further, the corpuscle responds to the vibration imposed on it, which is not necessarily the same as that imposed on the surface of the skin and may be filtered and transmitted differently because of structure or compositional differences in the tissues surrounding the corpuscle. Thus, two animals with corpuscles tuned to the same frequency might still experience very different optimal frequencies for detection because of other anatomical factors. The current study focused on the corpuscles proper, not on their location or the properties of the surrounding tissue, which would probably have considerable effect and could allow for further species-specific ranges for the sensory apparatus as a whole.

Animals with masses spanning over five orders of magnitude were analysed in this study. Despite such a large mass difference among the animals, lamellar corpuscle radii collected from histological images only spanned two orders of magnitude and did not show a strong dependence overall on animal size; for example, the whale PC is comparable in size to that of the rat. In some animals (e.g. the star-nosed mole), the operating size of the sensory appendage may in fact limit corpuscle size, but in other cases (e.g. the elephant), there is clearly enough room for a larger corpuscle. One interesting exception arises when considering the lamellar corpuscle radii collected in the mammalian class. When all mammalian corpuscles are analysed over the range of animal sizes (figure 1a), there is no dependence ($p = 0.71$) of lamellar corpuscle radius on animal size. When only terrestrial mammals are considered and aquatic mammals are removed from the analysis, there is a stronger dependence ($p = 0.16$) of corpuscle radius on animal size. The corpuscle radii for the two aquatic mammals analysed in this study (porpoise and whale) are smaller than their terrestrial counterparts of similar size. For example, the porpoise, which has a mass of 54.5 kg, has a PC outer radius of 18.6 µm. By contrast, the human, which has an average mass of 62.1 kg, has a PC outer radius of 190 µm. The outer radius for the lamellar corpuscle in the porpoise analysed in this study falls in the middle of the dimensions reported for porpoise corpuscles [37] These corpuscles were obtained from the dorsal right phonic lip of the porpoise. Likewise, the corpuscles analysed in the whale were obtained from the sinus cavity [38]. While the lamellar corpuscles analysed for the aquatic mammals were found in the rostrum, those for the terrestrial mammals were localized to the periphery [6,14,29,32,35,36] or internal organs [21], with the exception of the mole [34], which also has a small PC outer radius compared with the similarly sized mouse and rat. The differences in anatomical location between corpuscles found in aquatic and terrestrial mammals, and the functional differences that arise due to these locations, may account for the differences in lamellar corpuscle outer radius between the two groups. Additionally, variations in corpuscle biomechanical properties between species were not taken into account in these simulations, as these data are not available. Mechanical experiments have been performed on human PCs under steady-state conditions [29], but no biomechanical responses have been reported for corpuscles from other species or under dynamic conditions such as those experienced *in vivo*.

Overall, the results of this study show that animal size or class alone cannot be used to predict differences in corpuscle structure or peak frequency. These data provide insight into the effect of corpuscle size on the lamellar organization in the outer core. Specifically, corpuscle outer radius can predict the average lamellar thickness and the number of lamellae within that corpuscle. The dependence of lamellar number and thickness on corpuscle outer radius ($p < 0.07$), in contrast with their lack of dependence on animal mass ($p > 0.38$), suggests that the internal organization of a lamellar corpuscle is determined by the size of that corpuscle rather than the size of the corpuscle's animal host. Future work must be done to understand the physiological motivations behind species-specific frequency sensitivities in sensory end organs and to validate the model with experimental findings.

Tuning curves have been constructed from electrophysiological studies on animals such as the cat [3] and the duck [41]. In both animals, the peak frequency of the recorded corpuscles varies between structures (approx. 50–500 Hz in the duck bill [41]), but without reports on the size and lamellar organization of the tested corpuscles these variations in functional response cannot be compared with variations in corpuscle size or internal structure. Additionally, published micrographs were used to extract structural parameters for this study. The micrographs selected for publication are only representative samples of what is clearly a structurally and functionally diverse corpuscle population (figure 3 and its source of [41], figure 1 of [17], or the particularly thorough account of [5]). There may have been selection bias, such as preference for larger and thus more easily visible corpuscles. Despite these limitations, our simulated response for the duck HC based on a single micrograph [24] fell within the range of experimental results [41] (figure 3) obtained from seven different corpuscles. Those seven corpuscles showed a wide range of behaviours, which is perhaps not surprising given the high sensitivity to corpuscle radius, lamella thickness and especially the number of lamellae in the model. These parameters were not reported for the tested corpuscles [41], so we can only speculate that structural differences would account for the range of the experimental data in figure 3. We found previously [16] that the optimal frequency predicted by the model scales linearly with the corpuscle modulus and lamellar thickness and superlinearly ($N^{3.475}$) with the number of lamellae. For example, a 22% increase in the number of lamellae, which certainly seems possible from one corpuscle to another, would double the optimal frequency. It is also notable that the study of [41] involved stimulation of the duck's bill, not the isolated Herbst corpuscle, which would further contribute to the variation observed in figure 3.

We stress that our model for each species is of a single corpuscle, taken from a single published micrograph for that species and does not involve parameters such as location in the surrounding

tissue or interactions with other corpuscles [17]. All of these are significant factors and should be considered in future models, particularly if one were interested in a specific animal and had greater data available. This work illustrates broad trends in corpuscle response but will need further refinement for application to a specific species.

Additionally, the model used in this study assumes that the central neurite in each simulated corpuscle has the same structural and electrophysiological properties. The neurite properties used in this study were previously tuned to the functional response of the cat PC [3,15]. Fitting the neurite properties based on the morphology in published micrographs or the electrophysiological response of tuning curves when applicable would better capture variations between corpuscles in different species.

It must be recognized that the calculations in this study remain quite rudimentary. In addition to the lack of species-specific neurobiology, we did not account for the fact that the corpuscle may be located at different depths within the skin in different animals, and that the skin of these animals may have different mechanical properties and thus transmit vibrations differently. Previous work using this model has shown that the distance between the stimulating probe and a corpuscle embedded in skin affects both the amplitude and phase shift of the vibration transmitted to the receptor's core [17], so any changes in skin depth or mechanics would affect signal propagation through the tissue. Nevertheless, this work raises interesting questions about the convergence of lamellar corpuscle sensitivity across a wide range of animals and provides a survey of corpuscle structural information for use in future studies.

Ethics. No human or animal subjects were used in this study.

Data accessibility. Data has been uploaded as electronic supplementary material.

Authors' contribution. V.H.B. and J.C.Q.-H. conceived the study and analysed the data. E.T.B. and O.K.J. performed the literature search and made measurements. J.C.Q.-H. performed the computations. V.H.B. and J.C.Q.-H. did the primary writing of the manuscript. All authors reviewed and approved the final manuscript and agree to be held accountable for the work therein.

Competing interests. The authors declare no competing interests.

Funding. This work was supported by a University of Minnesota Doctoral Dissertation Fellowship to J.C.Q.-H.

Acknowledgements. This is not relevant to the work as everyone who has contributed to the study has met the authorship criteria.

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
