## [Reviewer comments · Royal Society Open Science]

Review History

RSOS-191439.R0 (Original submission)

Review form: Reviewer 1

Is the manuscript scientifically sound in its present form?

Yes

Are the interpretations and conclusions justified by the results?

Yes

Is the language acceptable?

Yes

Do you have any ethical concerns with this paper?

No

Have you any concerns about statistical analyses in this paper?

No

Recommendation?

Accept with minor revision (please list in comments)

Comments to the Author(s)

Review report on the manuscript entitled “An inter-species computational analysis of vibrotactile sensitivity in Pacinian and Herbst corpuscles” by Quindlen-Hotek et al.

In their very nicely written and well-presented manuscript, Quindlen-Hotek and coauthors report on a computational study of the neural responses of lamellar corpuscles to surface vibrations, for 19 different animal species. The authors simulate tuning curves (threshold amplitude versus frequency of the applied mechanical stimulus) using a model of onion-like corpuscles that they have published earlier. This model is based on a finite element mechanical model of both the inner and outer cores of the corpuscle, combined with a computational electrochemical model of the neurite’s membrane. To provide a comparison between all 19 species corpuscles, the authors rely on a literature-based image analysis of available micrographs to extract corpuscles structural parameters such as their size, their number and width of lamellae. Using these measured parameters allows them to exhibit that there is no correlation between their values and the animal mass, and that there exists a rather good correlation between the number of lamellae and the outer core radius the corpuscle. All simulated tuning curves have a characteristic band-pass like shape with an optimal frequency. But most importantly, the authors find that, even though there is a high variability in the values of the structural parameters, for almost all animals tested, the optimal frequency is centered around 40-50 Hz. This does not hold however for humans and geese, for which this frequency rather lies around 130-170 Hz. This result in itself is sufficiently intriguing and interesting from an evolutionary point of view, which would justify its publication.

Yet, I have a few questions and comments that I think need to be addressed first.

1 – On the technical side, my understanding is that the authors apply a homogeneous oscillating pressure on the corpuscle to calculate their tuning curve. How realistic is such a stimulus? I would have expected that *in vivo*, the applied stimulus is not necessarily spatially homogenous on the outer shell. How would the neural response change if stresses were applied locally on the corpuscle? Would it shift the optimum frequency?

2 – Since the authors’ model has never been compared to actual avian species, the authors calculate the tuning curve for the Herbst corpuscles of ducks for which electrophysiological data are available in Ref. 32. Surely, as the authors write, “[the simulated tuning curve] falls comfortably within the range of observed thresholds...”. Yet, one can immediately notice that all 7 experimental tuning curves are very different and yield potentially different optimum frequencies that vary from typically 50 Hz to 200 Hz. Unless I am mistaken, the striking difference is that the reported experiments were performed *in vivo* and mechanical stimulations were applied through the surrounding tissue (beak or skin) that are likely to contribute to the overall filtering properties. It is highly suspected indeed that the mechanical and topographical properties of the surrounding tissue could participate in the filtering process (see for instance Scheibert et al. *Science* 323, 1503, 2009 and Eastwood et al. *PNAS* 112, E6955-63, 2015). Could this explain the differences between the calculated tuning curve and the experimental ones, a fact that the authors themselves mention in their conclusion saying that “...we did not account for the fact that the corpuscle may be located at different depths within the skin in different animals...”? At this point, one can thus question whether or not the calculated optimum frequencies and sensitivity range would remain rather conserved through all tested species (except for humans and geese). Unless I am mistaken, in any case it seems to me that the authors have calculated an intrinsic frequency response of these corpuscles that is solely related to their inner structure and that does not consider the influence of the surrounding tissue. I feel that this point should be made much clearer in the manuscript.

Review form: Reviewer 2

Is the manuscript scientifically sound in its present form?

Yes

Are the interpretations and conclusions justified by the results?

Yes

Is the language acceptable?

Yes

Do you have any ethical concerns with this paper?

No

Have you any concerns about statistical analyses in this paper?

No

Recommendation?

Major revision is needed (please make suggestions in comments)

Comments to the Author(s)

This paper explains how the PC model developed earlier by the authors is useful in determining the frequency response and sensitivity of vibration sensation in nineteen different species. They conclude that the PC across different species of different sizes and structures to achieve similar frequency-detection bands.

The following points are major limitations of the paper:

- 1) The Table-1 could show ranges of the PCs geometry parameters, instead of mean value alone, for each species. It is well known that in each species the PC's size distribution is wide, outer diameter varies, the number of lamellae varies, thickness varies for different anatomical sites, also due to the age. It is not clear if the conclusion can be drawn without considering these variations. Perhaps, adding a range for each parameter for each species will help. For example, the outer most 5 to 7 perineural lamellae are closely spaced and densely packed with collagen fibrils in human PCs. Therefore, the average value alone may not serve the purpose. If the range can be ignored, the rationale for ignoring the variation can be added to the text.
- 2) The lamellar thickness and the clearance estimated from the microscopic slides may not be the exact functional interlamellar spacing as assumed in this paper. i) B. Munger et al., "A revaluation of the cytology of cat Pacinian corpuscles," *Cell Tissue Res.*, vol. 253, no. 1, pp. 83-93, 1988. ii) K. Sames et al., "Lectin and Proteoglycan Histochemistry of Feline Pacinian Corpuscles," *J. Histochem. Cytochem.*, vol. 49, no. 1, pp. 19-28, Jan. 2001. This assumption can be explicitly stated in the methodology.
- 3) It is not very clear if the number of lamellae reported in the paper includes the Inner core. It appears that the number represents only the outer core. If not included, reasons for reporting only the outer core could be mentioned. Otherwise, the inner core diameter, number of lamellae, and thickness of the cleft can be added to the text. Again, the range of each of these parameters can be added.
- 4) The tuning curve of the duck's Herbst corpuscle (HC) is simulated and shown in figure 3. The simulated tuning curve is validated using seven different experimental tuning curves. It is observed that the lower peak is shifted to the left compared to the literature data. The simulated tuning curve has an almost infinite slope. Also, the frequency range reported in the simulated curve was well short of the experimental frequency range, since there is no correlation between lamellae radii and animal size. The reasons for these observations are not discussed.

- 5) Also, it is well known that the tuning curve varies depending on the site, stimulation, and the procedure to measure. When listing each of the seven tuning curves, it may be useful to list the experimental parameters used for measurement.
- 6) It is not clear from the text if the tuning curve from the literature considered for validation corresponds to the structure of the PC listed in Table-1. That is, the authors can mention whether there is a correlation between Table-1 and figure-4.
- 7) Sensitivity analysis of the model can serve the purpose of this paper by listing those parameters that change the tuning curve most or least.
- 8) Although the authors have mentioned their difficulties in collecting the species-specific parameters, it would be useful to know the biomechanical properties of the corpuscles considered for the prediction of tuning curves.

Decision letter (RSOS-191439.R0)

28-Oct-2019

Dear Professor Quindlen-Hotek,

The editors assigned to your paper ("An inter-species computational analysis of vibrotactile sensitivity in Pacinian and Herbst corpuscles") have now received comments from reviewers. We would like you to revise your paper in accordance with the referee and Associate Editor suggestions which can be found below (not including confidential reports to the Editor). Please note this decision does not guarantee eventual acceptance.

Please submit a copy of your revised paper before 20-Nov-2019. Please note that the revision deadline will expire at 00.00am on this date. If we do not hear from you within this time then it will be assumed that the paper has been withdrawn. In exceptional circumstances, extensions may be possible if agreed with the Editorial Office in advance. We do not allow multiple rounds of revision so we urge you to make every effort to fully address all of the comments at this stage. If deemed necessary by the Editors, your manuscript will be sent back to one or more of the original reviewers for assessment. If the original reviewers are not available, we may invite new reviewers.

- Data accessibility

<http://datadryad.org/submit?journalID=RSOS&manu=RSOS-191439>

- Competing interests

- Authors' contributions

- Acknowledgements

- Funding statement

Kind regards,
Anita Kristiansen

Editorial Coordinator
 Royal Society Open Science
 openscience@royalsociety.org

on behalf of Professor Madhusudhan Venkadesan (Associate Editor) and R. Kerry Rowe (Subject Editor)
 openscience@royalsociety.org

Associate Editor's comments (Professor Madhusudhan Venkadesan):
 Comments to the Author:

Dear authors. As you can see, the reviewers are supportive but have some major questions that need addressing. The central concern appears to be one of applicability of the model to real Pacinian corpuscles. The concerns include modeling assumptions (multi-layered structure, surrounding tissue, biomechanical properties of the tissue), the lack of any sensitivity analysis, and the lack of insights through analysis of the models. The main result may be viewed more as a hypothesis that is generated from simulations of a specific model. Experimental agreement in the future may indicate that the model merits further analysis, or disagreement may indicate that some of the assumptions that went into building the model are broken. Seen through this lens, the paper may be strengthened by expanding the level and depth of analysis of the model. The hypothesis is intriguing, that there is somehow an invariance across species. Some of the novelty of the hypothesis could be diluted because there is substantial variation across experimental measurements, which is not considered by the authors.

Reviewers' Comments to Author:
 Reviewer: 1

Comments to the Author(s)

Review report on the manuscript entitled "An inter-species computational analysis of vibrotactile sensitivity in Pacinian and Herbst corpuscles" by Quindlen-Hotek et al.

In their very nicely written and well-presented manuscript, Quindlen-Hotek and coauthors report on a computational study of the neural responses of lamellar corpuscles to surface vibrations, for 19 different animal species. The authors simulate tuning curves (threshold amplitude versus frequency of the applied mechanical stimulus) using a model of onion-like corpuscles that they have published earlier. This model is based on a finite element mechanical model of both the inner and outer cores of the corpuscle, combined with a computational electrochemical model of the neurite's membrane. To provide a comparison between all 19 species corpuscles, the authors rely on a literature-based image analysis of available micrographs to extract corpuscles structural parameters such as their size, their number and width of lamellae. Using these measured parameters allows them to exhibit that there is no correlation between their values and the animal mass, and that there exists a rather good correlation between the number of lamellae and the outer core radius the corpuscle. All simulated tuning curves have a characteristic band-pass like shape with an optimal frequency. But most importantly, the authors find that, even though there is a high variability in the values of the structural parameters, for almost all animals tested, the optimal frequency is centered around 40-50 Hz. This does not hold however for humans and geese, for which this frequency rather lies around 130-170 Hz. This result in itself is sufficiently intriguing and interesting from an evolutionary point of view, which would justify its publication.

Yet, I have a few questions and comments that I think need to be addressed first.

1 - On the technical side, my understanding is that the authors apply a homogeneous oscillating pressure on the corpuscle to calculate their tuning curve. How realistic is such a stimulus? I would have expected that in vivo, the applied stimulus is not necessarily spatially homogenous

on the outer shell. How would the neural response change if stresses were applied locally on the corpuscle? Would it shift the optimum frequency?

2 – Since the authors' model has never been compared to actual avian species, the authors calculate the tuning curve for the Herbst corpuscles of ducks for which electrophysiological data are available in Ref. 32. Surely, as the authors write, "[the simulated tuning curve] falls comfortably within the range of observed thresholds...". Yet, one can immediately notice that all 7 experimental tuning curves are very different and yield potentially different optimum frequencies that vary from typically 50 Hz to 200 Hz. Unless I am mistaken, the striking difference is that the reported experiments were performed *in vivo* and mechanical stimulations were applied through the surrounding tissue (beak or skin) that are likely to contribute to the overall filtering properties. It is highly suspected indeed that the mechanical and topographical properties of the surrounding tissue could participate in the filtering process (see for instance Scheibert et al. *Science* 323, 1503, 2009 and Eastwood et al. *PNAS* 112, E6955-63, 2015). Could this explain the differences between the calculated tuning curve and the experimental ones, a fact that the authors themselves mention in their conclusion saying that "...we did not account for the fact that the corpuscle may be located at different depths within the skin in different animals..."? At this point, one can thus question whether or not the calculated optimum frequencies and sensitivity range would remain rather conserved through all tested species (except for humans and geese). Unless I am mistaken, in any case it seems to me that the authors have calculated an intrinsic frequency response of these corpuscles that is solely related to their inner structure and that does not consider the influence of the surrounding tissue. I feel that this point should be made much clearer in the manuscript.

Reviewer: 2

Comments to the Author(s)

This paper explains how the PC model developed earlier by the authors is useful in determining the frequency response and sensitivity of vibration sensation in nineteen different species. They conclude that the PC across different species of different sizes and structures to achieve similar frequency-detection bands.

The following points are major limitations of the paper:

- 1) The Table-1 could show ranges of the PCs geometry parameters, instead of mean value alone, for each species. It is well known that in each species the PC's size distribution is wide, outer diameter varies, the number of lamellae varies, thickness varies for different anatomical sites, also due to the age. It is not clear if the conclusion can be drawn without considering these variations. Perhaps, adding a range for each parameter for each species will help. For example, the outer most 5 to 7 perineural lamellae are closely spaced and densely packed with collagen fibrils in human PCs. Therefore, the average value alone may not serve the purpose. If the range can be ignored, the rationale for ignoring the variation can be added to the text.
- 2) The lamellar thickness and the clearance estimated from the microscopic slides may not be the exact functional interlamellar spacing as assumed in this paper. i) B. Munger et al., "A reevaluation of the cytology of cat Pacinian corpuscles," *Cell Tissue Res.*, vol. 253, no. 1, pp. 83-93, 1988. ii) K. Sames et al., "Lectin and Proteoglycan Histochemistry of Feline Pacinian Corpuscles," *J. Histochem. Cytochem.*, vol. 49, no. 1, pp. 19-28, Jan. 2001. This assumption can be explicitly stated in the methodology.
- 3) It is not very clear if the number of lamellae reported in the paper includes the Inner core. It appears that the number represents only the outer core. If not included, reasons for reporting only the outer core could be mentioned. Otherwise, the inner core diameter, number of lamellae, and thickness of the cleft can be added to the text. Again, the range of each of these parameters can be added.

- 4) The tuning curve of the duck's Herbst corpuscle (HC) is simulated and shown in figure 3. The simulated tuning curve is validated using seven different experimental tuning curves. It is observed that the lower peak is shifted to the left compared to the literature data. The simulated tuning curve has an almost infinite slope. Also, the frequency range reported in the simulated curve was well short of the experimental frequency range, since there is no correlation between lamellae radii and animal size. The reasons for these observations are not discussed.
- 5) Also, it is well known that the tuning curve varies depending on the site, stimulation, and the procedure to measure. When listing each of the seven tuning curves, it may be useful to list the experimental parameters used for measurement.
- 6) It is not clear from the text if the tuning curve from the literature considered for validation corresponds to the structure of the PC listed in Table-1. That is, the authors can mention whether there is a correlation between Table-1 and figure-4.
- 7) Sensitivity analysis of the model can serve the purpose of this paper by listing those parameters that change the tuning curve most or least.
- 8) Although the authors have mentioned their difficulties in collecting the species-specific parameters, it would be useful to know the biomechanical properties of the corpuscles considered for the prediction of tuning curves.

Author's Response to Decision Letter for (RSOS-191439.R0)

See Appendix A.

RSOS-191439.R1 (Revision)

Review form: Reviewer 1

Is the manuscript scientifically sound in its present form?

Yes

Are the interpretations and conclusions justified by the results?

Yes

Is the language acceptable?

Yes

Do you have any ethical concerns with this paper?

No

Have you any concerns about statistical analyses in this paper?

No

Recommendation?

Accept as is

Comments to the Author(s)

The authors have addressed all points that I raised. I am pleased with their answers and the modifications they provided to their manuscript. I therefore recommend the manuscript for publication as it is.

Review form: Reviewer 2**Is the manuscript scientifically sound in its present form?**

No

Are the interpretations and conclusions justified by the results?

No

Is the language acceptable?

Yes

Do you have any ethical concerns with this paper?

No

Have you any concerns about statistical analyses in this paper?

No

Recommendation?

Major revision is needed (please make suggestions in comments)

Comments to the Author(s)

The authors have not addressed main concerns that were raised in the last review.

The claim of invariance across species needs stronger evidence. Every extraordinary hypothesis needs extraordinary proof or validation.

1) The experimental evidence shows substantial variation in the PC data of different species, and the authors have failed to explain the difference between the experiment and simulation results in the rebuttal, specifically in the figure labelled "Comparison of simulated tuning curve for the duck Herbst corpuscle to published electrophysiological data [35] collected from Herbst corpuscles". The simulated data are within the experimentally reported range but show some considerable differences. The justification for the frequency response such as the restriction below 200Hz is not clear. The threshold of the corpuscle-only simulation should be better than any of the recorded data. However, at least three of the recorded data have better sensitivity than the simulated one.

2) Is it enough to show that the reported simulation results falls somewhere in the range of the experimental data, but not explaining the entire experimental data? The only data for the validation is from duck HC, that too the validation of the simulated data with the experimental is not matching the range, and therefore oversimplification.

3) In the rebuttal, the authors claim that the experiments did not account for in the corpuscle-only simulations. Rather, it should be otherway, the corpuscle-only simulation did not match with the experimental data. Invariance of corpuscle-only simulation need to be established better.

4) The authors agree that the corpuscle may be located at different depths within the skin in different animals. More importantly, the biomechanics of the skin of different animals may be

different. How changes in the skin properties affect the optimal frequency of the PC could be added to the text to support the claim.

5) The literature for the experimental data quoted in the paper do not have enough details such as the stimulation site, probe, method, frequency etc. I am not sure if such incomplete data can be used for comparison with the simulated data. If used, it is too much of a stretch.

6) In spite of so much of variation of PC size, even within a species, for most of the species analyzed in this paper, experimental data from only one published micrograph is used. What is the confidence that this micrograph is a representative data within the species?

The claim is very interesting, however it needs more supporting evidences.

Decision letter (RSOS-191439.R1)

03-Jan-2020

Dear Professor Quindlen-Hotek:

Manuscript ID RSOS-191439.R1 entitled "An inter-species computational analysis of vibrotactile sensitivity in Pacinian and Herbst corpuscles" which you submitted to Royal Society Open Science, has been reviewed. The comments of the reviewer(s) are included at the bottom of this letter.

Please submit a copy of your revised paper before 26-Jan-2020. Please note that the revision deadline will expire at 00.00am on this date. If we do not hear from you within this time then it will be assumed that the paper has been withdrawn. In exceptional circumstances, extensions may be possible if agreed with the Editorial Office in advance.

While we do not generally allow multiple rounds of revision, the Editors have granted you a further opportunity to revise your paper. We urge you to make every effort to fully address all of the comments at this stage. If deemed necessary by the Editors, your manuscript will be sent back to one or more of the original reviewers for assessment. If the original reviewers are not available we may invite new reviewers. If you do not satisfy the Editors and reviewers the paper is ready for acceptance following this revision, it may be rejected.

- Ethics statement

- Data accessibility

- Competing interests

- Authors' contributions

- Acknowledgements

- Funding statement

Kind regards,

Andrew Dunn

on behalf of Professor Madhusudhan Venkadesan (Associate Editor) and R. Kerry Rowe (Subject Editor)
 openscience@royalsociety.org

Associate Editor Comments to Author (Professor Madhusudhan Venkadesan):

As seen from one of the referees, the authors have not truly addressed the critical comments. It is important that the authors should fully address this referee's comments in their revision. I cannot sufficiently emphasize the importance of addressing all the referees, however critical they may be of your work.

Reviewer comments to Author:

Reviewer: 1

Comments to the Author(s)

The authors have addressed all points that I raised. I am pleased with their answers and the modifications they provided to their manuscript. I therefore recommend the manuscript for publication as it is.

Reviewer: 2

Comments to the Author(s)

The authors have not addressed main concerns that were raised in the last review.

The claim of invariance across species needs stronger evidence. Every extraordinary hypothesis needs extraordinary proof or validation.

1) The experimental evidence shows substantial variation in the PC data of different species, and the authors have failed to explain the difference between the experiment and simulation results in the rebuttal, specifically in the figure labelled "Comparison of simulated tuning curve for the duck Herbst corpuscle to published electrophysiological data [35] collected from Herbst corpuscles". The simulated data are within the experimentally reported range but show some considerable differences. The justification for the frequency response such as the restriction below 200Hz is not clear. The threshold of the corpuscle-only simulation should be better than any of the recorded data. However, at least three of the recorded data have better sensitivity than the simulated one.

2) Is it enough to show that the reported simulation results falls somewhere in the range of the experimental data, but not explaining the entire experimental data? The only data for the validation is from duck HC, that too the validation of the simulated data with the experimental is not matching the range, and therefore oversimplification.

3) In the rebuttal, the authors claim that the experiments did not account for in the corpuscle-only simulations. Rather, it should be otherway, the corpuscle-only simulation did not match with the experimental data. Invariance of corpuscle-only simulation need to be established better.

4) The authors agree that the corpuscle may be located at different depths within the skin in different animals. More importantly, the biomechanics of the skin of different animals may be different. How changes in the skin properties affect the optimal frequency of the PC could be added to the text to support the claim.

5) The literature for the experimental data quoted in the paper do not have enough details such as

the stimulation site, probe, method, frequency etc. I am not sure if such incomplete data can be used for comparison with the simulated data. If used, it is too much of a stretch.

6) In spite of so much of variation of PC size, even within a species, for most of the species analyzed in this paper, experimental data from only one published micrograph is used. What is the confidence that this micrograph is a representative data within the species?

The claim is very interesting, however it needs more supporting evidences.

Author's Response to Decision Letter for (RSOS-191439.R1)

See Appendix B.

Decision letter (RSOS-191439.R2)

06-Mar-2020

Dear Professor Quindlen-Hotek:

On behalf of the Editors, I am pleased to inform you that your Manuscript RSOS-191439.R2 entitled "An inter-species computational analysis of vibrotactile sensitivity in Pacinian and Herbst corpuscles" has been accepted for publication in Royal Society Open Science subject to minor revision in accordance with the referee suggestions. Please find the referees' comments at the end of this email.

The reviewers and Subject Editor have recommended publication, but also suggest some minor revisions to your manuscript. Therefore, I invite you to respond to the comments and revise your manuscript.

- Ethics statement

- Data accessibility

<http://datadryad.org/submit?journalID=RSOS&manu=RSOS-191439.R2>

- **Competing interests**

- **Authors' contributions**

- **Acknowledgements**

- **Funding statement**

Because the schedule for publication is very tight, it is a condition of publication that you submit the revised version of your manuscript before 15-Mar-2020. Please note that the revision deadline will expire at 00.00am on this date. If you do not think you will be able to meet this date please let me know immediately.

1) A text file of the manuscript (tex, txt, rtf, docx or doc), references, tables (including captions) and figure captions. Do not upload a PDF as your "Main Document".

- 2) A separate electronic file of each figure (EPS or print-quality PDF preferred (either format should be produced directly from original creation package), or original software format)
- 3) Included a 100 word media summary of your paper when requested at submission. Please ensure you have entered correct contact details (email, institution and telephone) in your user account
- 4) Included the raw data to support the claims made in your paper. You can either include your data as electronic supplementary material or upload to a repository and include the relevant doi within your manuscript
- 5) All supplementary materials accompanying an accepted article will be treated as in their final form. Note that the Royal Society will neither edit nor typeset supplementary material and it will be hosted as provided. Please ensure that the supplementary material includes the paper details where possible (authors, article title, journal name).

on behalf of Professor Madhusudhan Venkadesan (Associate Editor) and R. Kerry Rowe (Subject Editor)
openscience@royalsociety.org

Associate Editor Comments to Author (Professor Madhusudhan Venkadesan):

Comments to the Author:

The responses are satisfactory but the claim in the last sentence of the abstract still concerns me. Given the lack of direct experimental evidence and missing features such as the tissues surrounding the corpuscles, the conclusion of this paper may only be stated as a numerically generated hypothesis. The current concluding sentence of the abstract states the hypothesis as if it's a proven fact. Such a statement runs the risk of future papers claiming that as a result of this work, without appropriate caution. It is important that the claim should be tempered to state that the inter-species conclusion is truly a hypothesis generated by the computational modeling.

Author's Response to Decision Letter for (RSOS-191439.R2)

See Appendix C.

Decision letter (RSOS-191439.R3)

19-Mar-2020

Dear Professor Quindlen-Hotek,

It is a pleasure to accept your manuscript entitled "An inter-species computational analysis of vibrotactile sensitivity in Pacinian and Herbst corpuscles" in its current form for publication in Royal Society Open Science. The comments of the reviewer(s) who reviewed your manuscript are included at the foot of this letter.

on behalf of Professor Madhusudhan Venkadesan (Associate Editor) and R. Kerry Rowe (Subject Editor)
openscience@royalsociety.org

Appendix A

We thank the reviewers for their valuable comments. In our response below, the reviewers' comments are given in **boldface**, followed by our response in plain font. Changes to the text are *italicized* below and highlighted in **yellow** in the manuscript.

Comments to the Author:

Dear authors. As you can see, the reviewers are supportive but have some major questions that need addressing. The central concern appears to be one of applicability of the model to real Pacinian corpuscles. The concerns include modeling assumptions (multi-layered structure, surrounding tissue, biomechanical properties of the tissue), the lack of any sensitivity analysis, and the lack of insights through analysis of the models. The main result may be viewed more as a hypothesis that is generated from simulations of a specific model. Experimental agreement in the future may indicate that the model merits further analysis, or disagreement may indicate that some of the assumptions that went into building the model are broken. Seen through this lens, the paper may be strengthened by expanding the level and depth of analysis of the model. The hypothesis is intriguing, that there is somehow an invariance across species. Some of the novelty of the hypothesis could be diluted because there is substantial variation across experimental measurements, which is not considered by the authors.

Reviewers' Comments to Author:

Reviewer: 1

Comments to the Author(s)

Review report on the manuscript entitled "An inter-species computational analysis of vibrotactile sensitivity in Pacinian and Herbst corpuscles" by Quindlen-Hotek et al.

In their very nicely written and well-presented manuscript, Quindlen-Hotek and coauthors report on a computational study of the neural responses of lamellar corpuscles to surface vibrations, for 19 different animal species. The authors simulate tuning curves (threshold amplitude versus frequency of the applied mechanical stimulus) using a model of onion-like corpuscles that they have published earlier. This model is based on a finite element mechanical model of both the inner and outer cores of the corpuscle, combined with a computational electrochemical model of the neurite's membrane. To provide a comparison between all 19 species corpuscles, the authors rely on a literature-based image analysis of available micrographs to extract corpuscles structural parameters such as their size, their number and width of lamellae. Using these measured parameters allows them to exhibit that there is no correlation between their values and the animal mass, and that there exists a rather good correlation between the number of lamellae and the outer core radius the corpuscle. All simulated tuning curves have a characteristic band-pass like shape with an optimal frequency. But most importantly, the authors find that, even though there is a high variability in the values of the structural parameters, for almost all animals tested, the optimal frequency is centered around 40-50 Hz. This does not hold however for humans and geese, for which this frequency rather lies around 130-170 Hz. This result in itself is sufficiently intriguing and interesting from

an evolutionary point of view, which would justify its publication.

Yet, I have a few questions and comments that I think need to be addressed first.

1 – On the technical side, my understanding is that the authors apply a homogeneous oscillating pressure on the corpuscle to calculate their tuning curve. How realistic is such a stimulus? I would have expected that in vivo, the applied stimulus is not necessarily spatially homogenous on the outer shell. How would the neural response change if stresses were applied locally on the corpuscle? Would it shift the optimum frequency?

The oscillating pressure is not homogeneous around the surface (such a stimulus would do almost nothing because of the incompressibility of the tissue). Rather, it is described by a sinusoid with peaks at the pole and equator of the spherical corpuscle (see eqn 31 of [15] for details). We have modified the following sentence in the Methods section to clarify this point:

The outermost shell of the Stage 1 model is vibrated with a spatiotemporal pressure wave whose peaks occur at the poles and equator of the corpuscle and whose temporal frequency is specified (see equation 31 of [15] for more detail).

The stimulus is meant to simulate compression of an isolated corpuscle, as would be used in electrophysiological experiments such as those published in Bolanowski & Zwislocki 1984 [3 in the paper]. In vivo, the stimulus is applied to the surface of the skin or tissue surrounding the corpuscle and the neural response of the corpuscle will depend on factors such as the corpuscle's depth and surrounding tissue, as shown in Quindlen-Hotek & Barocas 2018 [17 in the paper], as the reviewer suggested.

2 – Since the authors' model has never been compared to actual avian species, the authors calculate the tuning curve for the Herbst corpuscles of ducks for which electrophysiological data are available in Ref. 32. Surely, as the authors write, “[the simulated tuning curve] falls comfortably within the range of observed thresholds...”. Yet, one can immediately notice that all 7 experimental tuning curves are very different and yield potentially different optimum frequencies that vary from typically 50 Hz to 200 Hz. Unless I am mistaken, the striking difference is that the reported experiments were performed in vivo and mechanical stimulations were applied through the surrounding tissue (beak or skin) that are likely to contribute to the overall filtering properties. It is highly suspected indeed that the mechanical and topographical properties of the surrounding tissue could participate in the filtering process (see for instance Scheibert et al. Science 323, 1503, 2009 and Eastwood et al. PNAS 112, E6955-63, 2015). Could this explain the differences between the calculated tuning curve and the experimental ones, a fact that the authors themselves mention in their conclusion saying that “...we did not account for the fact that the corpuscle may be located at different depths within the skin in different animals...”? At this point, one can thus question whether or not the calculated optimum frequencies and sensitivity range would remain rather conserved through all tested species (except for humans and geese). Unless I am mistaken, in any case it seems to me that the authors have calculated an intrinsic frequency response of these corpuscles that is solely related to their inner structure and that does not consider the influence of the surrounding tissue. I feel that this point should be made much

clearer in the manuscript.

The reviewer is correct and raises an important consideration. We have modified the text in two ways in response. First, regarding the spread of the tuning curves in Figure 3, we have added the following paragraph to the Discussion sections, including the sentence underlined here to clarify that the experimental data were from the whole bill, not the isolated corpuscle.

Despite these limitations, our simulated response for the duck HC based on a single micrograph [24] fell within the range of experimental results [35] (Figure 3) obtained from 7 different corpuscles. Those seven corpuscles showed a wide range of behaviors, which is perhaps not surprising given the high sensitivity to corpuscle radius, lamella thickness, and especially number of lamellae in the model. We found previously [16] that the optimal frequency predicted by the model scales linearly with the corpuscle modulus and lamellar thickness and superlinearly ($N^{3.475}$) with the number of lamellae. For example, a 22% increase in the number of lamellae, which certainly seems possible from one corpuscle to another, would double the optimal frequency. It is also notable that the study of [35] involved stimulation of the duck's bill, not the isolated Herbst corpuscle, which would further contribute to the variation observed in Figure 3.

Second, we added the following text to the Discussion section about the potential role of extracorporeal tissue:

Further, the corpuscle responds to the vibration imposed on it, which is not necessarily the same as that imposed on the surface of the skin and may be filtered and transmitted differently because of structure or compositional differences in the tissues surrounding the corpuscle. Thus, two animals with corpuscles tuned to the same frequency might still experience very different optimal frequencies for detection because of other anatomical factors. The current study focused on the corpuscles proper, not on their location or the properties of the surrounding tissue, which would likely have considerable effect and could allow for further species-specific ranges for the sensory apparatus as a whole.

Reviewer: 2

Comments to the Author(s)

This paper explains how the PC model developed earlier by the authors is useful in determining the frequency response and sensitivity of vibration sensation in nineteen different species. They conclude that the PC across different species of different sizes and structures to achieve similar frequency-detection bands.

The following points are major limitations of the paper:

1) The Table-1 could show ranges of the PCs geometry parameters, instead of mean value alone, for each species. It is well known that in each species the PC's size distribution is wide, outer diameter varies, the number of lamellae varies, thickness varies for different anatomical sites, also due to the age. It is not clear if the conclusion can be drawn without considering these variations. Perhaps, adding a range for each parameter for each species will help. For example, the outermost 5 to 7 perineural lamellae are closely spaced and densely packed with collagen fibrils in human PCs.

Therefore, the average value alone may not serve the purpose. If the range can be ignored, the rationale for ignoring the variation can be added to the text.

The reviewer is correct. A PC or HC's size, shape, diameter, and inner lamellar organization are widely varied across a single species. While this has been reported for some species like the cat, ranges for these parameters have not been reported for all species analyzed in this study. For most of the species analyzed here, publications included only one published micrograph, which we then analyzed in ImageJ.

We added the following text to the Methods to clarify this point:

In most cases, a diameter range was not reported by the paper in question, so all other species' structural properties were measured by analysis of histological images using ImageJ [25,26]. In some cases, only one histological image of the corpuscle was provided; in cases where the paper in question reported multiple images, the image with the clearest lamellar organization was chosen for analysis.

We also added the following text to the Results:

Table 1 contain a single value, and not a range, for each parameter reported for different species. While anatomical variation exists between corpuscles in a single species and has been reported in the literature [21,30–32], these detailed evaluations of corpuscle structure and published ranges have mainly been reported for the cat corpuscle. Therefore, for animal species other than the cat and duck, a single histological image was analyzed and the parameters determined from that one image are reported in Table 1.

2) The lamellar thickness and the clearance estimated from the microscopic slides may not be the exact functional interlamellar spacing as assumed in this paper. i) B. Munger et al., "A revaluation of the cytology of cat Pacinian corpuscles," Cell Tissue Res., vol. 253, no. 1, pp. 83–93, 1988. ii) K. Sames et al., "Lectin and Proteoglycan Histochemistry of Feline Pacinian Corpuscles," J. Histochem. Cytochem., vol. 49, no. 1, pp. 19–28, Jan. 2001. This assumption can be explicitly stated in the methodology.

The following text was added to the methods:

The lamellar thicknesses estimated in this approach depended on the resolution of the provided image and did not include factors such as the connective tissues, collagen fibrils, and proteoglycans present between lamellae [27,28].

3) It is not very clear if the number of lamellae reported in the paper includes the Inner core. It appears that the number represents only the outer core. If not included, reasons for reporting only the outer core could be mentioned. Otherwise, the inner core diameter, number of lamellae, and thickness of the cleft can be added to the text. Again, the range of each of these parameters can be added.

The inner core was modeled separately, so only outer core lamellae were counted. This has been clarified in the Methods section. The inner core's structure was not analyzed, and was modeled as a homogeneous material in COMSOL, as first published in Quindlen et al., 2016.

4) The tuning curve of the duck's Herbst corpuscle (HC) is simulated and shown in

figure 3. The simulated tuning curve is validated using seven different experimental tuning curves. It is observed that the lower peak is shifted to the left compared to the literature data. The simulated tuning curve has an almost infinite slope. Also, the frequency range reported in the simulated curve was well short of the experimental frequency range, since there is no correlation between lamellae radii and animal size. The reasons for these observations are not discussed.

The reviewer is correct that the model data are within the experimentally reported range but show some considerable differences. We have therefore added the following text to the Results section:

The computational curve also has much steeper slopes at the boundaries of the receptive range than the experimental curves, possibly due to an effect from the surrounding tissue in the experiments that was not accounted for in the corpuscle-only simulations. In particular, the distance between the stimulus and the corpuscle could not be controlled in Gregory's experiments, leading to potential attenuation of signal [17,36].

as well as additional text in the Discussion section as described following the reviewer's next comment.

5) Also, it is well known that the tuning curve varies depending on the site, stimulation, and the procedure to measure. When listing each of the seven tuning curves, it may be useful to list the experimental parameters used for measurement.

The reviewer raises an excellent point. Unfortunately, Gregory [32] did not report where the different stimuli were applied, so we cannot provide additional information. We have added the following sentence to emphasize that stimulating the bill is not the same as stimulating the Herbst corpuscle itself, as part of a larger discussion of variability (see response to comment 7 below):

It is also notable that the study of [35] involved stimulation of the duck's bill, not the isolated Herbst corpuscle, which would further contribute to the variation observed in Figure 3.

6) It is not clear from the text if the tuning curve from the literature considered for validation corresponds to the structure of the PC listed in Table-1. That is, the authors can mention whether there is a correlation between Table-1 and figure-4.

The two are unrelated, having been performed many years apart by different groups. The tuning curve study [35] did not report any structural data for mechanoreceptors and provided only neurophysiological data, so a direct comparison between structure and function was not possible. We have added text to the Methods section on this point, reading now as follows:

The simulated tuning curve of the duck corpuscle was compared (Figure 3) to published electrophysiological tuning curves from 7 HCs [35], with the caveat that the neurophysiological and structural studies were performed years apart and on different individual animals; unfortunately, the neurophysiological study [35] included no structural data, and the structural studies [23, 24] included no neurophysiological data.

7) Sensitivity analysis of the model can serve the purpose of this paper by listing those parameters that change the tuning curve most or least.

We previously (Quindlen *et al.*, 2017; ref 16 in the revised manuscript) conducted a thorough parametric study of the model. In that study, we used dimensional analysis arguments to conclude that the frequency of peak sensitivity should be proportional to the ratio ($Eh / \mu R_0$), and to the number of lamellae raised to the 3.475 power. Thus, doubling, say, the thickness h of the lamellae in the model would be expected roughly to double the optimal frequency, but doubling the number of lamellae would result in an 11-fold increase if nothing else were changed. The correlation was not perfect, and the range of our earlier study did not encompass the range of properties found in the current study, but it provides a basis for discussing the current results. We have added the following text to the Discussion section of the revised manuscript (the first sentence is unchanged from the original):

Despite these limitations, our simulated response for the duck HC based on a single micrograph [24] fell within the range of experimental results [35] (Figure 3) obtained from 7 different corpuscles. Those seven corpuscles showed a wide range of behaviors, which is perhaps not surprising given the high sensitivity to corpuscle radius, lamella thickness, and especially number of lamellae in the model. We found previously [16] that the optimal frequency predicted by the model scales linearly with the corpuscle modulus and lamellar thickness and superlinearly ($N^{3.475}$) with the number of lamellae. For example, a 22% increase in the number of lamellae, which certainly seems possible from one corpuscle to another, would double the optimal frequency. It is also notable that the study of [35] involved stimulation of the duck's bill, not the isolated Herbst corpuscle, which would further contribute to the variation observed in Figure 3.

8) Although the authors have mentioned their difficulties in collecting the species-specific parameters, it would be useful to know the biomechanical properties of the corpuscles considered for the prediction of tuning curves.

The Young's modulus used in these simulations was previously published from our group (Quindlen *et al.*, 2017, [29] in this paper) and determined via steady-state experiments performed on human corpuscles. No other studies have published biomechanical properties for corpuscles from any other species. Our 2017 paper is the only study that reports a Young's modulus for the Pacinian corpuscle.

Our lab is currently undertaking biomechanical experiments determine the corpuscle's mechanical properties under dynamic conditions on a physiologically-relevant time scale. While these mechanical properties will better inform models of the PC's biomechanical response to vibration, they are not yet available and thus the modulus published in [29] was used.

The following text was added to the Discussion to address the need for these data from different species:

Additionally, variations in corpuscle biomechanical properties between species were not taken into account in these simulations, as these data are not available. Mechanical experiments have been performed on human PCs under steady-state conditions [29], but no biomechanical responses have been reported for corpuscles from other species or under dynamic conditions such as those experienced in vivo.

Appendix B

We thank the reviewer for his/her additional comments. In our response below, the reviewer's comments are given in **boldface**, followed by our response in plain font. Changes to the text are *italicized* below and highlighted in **yellow** in the manuscript.

Reviewer: 1

Comments to the Author(s)

The authors have addressed all points that I raised. I am pleased with their answers and the modifications they provided to their manuscript. I therefore recommend the manuscript for publication as it is.

Reviewer: 2

Comments to the Author(s)

The authors have not addressed main concerns that were raised in the last review.

The claim of invariance across species needs stronger evidence. Every extraordinary hypothesis needs extraordinary proof or validation.

1) The experimental evidence shows substantial variation in the PC data of different species, and the authors have failed to explain the difference between the experiment and simulation results in the rebuttal, specifically in the figure labelled "Comparison of simulated tuning curve for the duck Herbst corpuscle to published electrophysiological data [35] collected from Herbst corpuscles". The simulated data are within the experimentally reported range but show some considerable differences. The justification for the frequency response such as the restriction below 200Hz is not clear. The threshold of the corpuscle-only simulation should be better than any of the recorded data. However, at least three of the recorded data have better sensitivity than the simulated one.

The goal for the simulated data in Figure 3 was to determine the peak frequency of a corpuscle with the structural parameters determined from published studies by Quilliam (1966) and Berkhoudt (1980). The tuning curve for the computational corpuscle contains values less than ~200 Hz because the goal was to construct a representative tuning curve that measured the peak frequency of a representative Herbst corpuscle in a duck bill. There is not a restriction as threshold amplitudes could be obtained for frequencies above 200 Hz. Amplitudes are reported for frequencies between 10 and ~200Hz to show a U-shaped curve.

The modeled corpuscle is one of many corpuscles that would be found within a single animal. Of all of the corpuscles located within a duck's bill, these corpuscles could be structurally different and some could be more sensitive. Additionally, our simulation was an isolated corpuscle, while the published experiment could be data from multiple corpuscles under a single stimulus which could lead to greater sensitivity.

The reviewer is correct that at least three of the experimentally-tested HCs may have peak frequency's above that of the simulated HC, but as we stated in the manuscript, we do not know the physical properties or depth of the experimentally-tested HCs and can only speculate as to why these tested HCs may be optimized to higher frequencies. To address this, the following text was added to the Results:

Furthermore, the physical properties and in vivo depths for the HCs tested by Gregory were not reported [35], and knowledge of these parameters would provide insight into the experimentally-obtained tuning curves how they compare to the simulated response.

2) Is it enough to show that the reported simulation results falls somewhere in the range of the experimental data, but not explaining the entire experimental data? The only data for the validation is from duck HC, that too the validation of the simulated data with the experimental is not matching the range, and therefore oversimplification.

The reviewer is correct that the only data used for validation is from the duck HC, but this is because that is the only experimental data available at this moment. If more experimental data were available, we would include it in this validation. Because the paper by Gregory with the original data does not contain information that would explain the range of experimental results (i.e corpuscle properties, corpuscle depth), we can only hypothesize as to why the seven tested HCs show such a wide range of behaviors. To address this concern, we have added the following text to the Discussion:

These parameters were not reported for the tested corpuscles [35], so we can only speculate that structural differences would account for the range of the experimental data in Figure 3.

Additionally, we added the following text to the Discussion to account for the fact that simulated data were only taken from one published image:

We stress that our model for each species is of a single corpuscle, taken from a single published micrograph for that species, and does not involve parameters such as location in the surrounding tissue or interactions with other corpuscles [17]. All of these are significant factors and should be considered in future models, particularly if one were interested in a specific animal and had greater data available. This work illustrates broad trends in corpuscle response but will need further refinement for application to a specific species.

3) In the rebuttal, the authors claim that the experiments did not account for in the corpuscle-only simulations. Rather, it should be otherway, the corpuscle-only simulation did not match with the experimental data. Invariance of corpuscle-only simulation need to be established better.

We think that the reviewer's concerns are addressed in the following section of Results, which states that the corpuscle-only simulations did not match the experimental data. This text was added to the manuscript in the last round of revisions:

The computational curve also has much steeper slopes at the boundaries of the receptive range than the experimental curves, possibly due to an effect from the surrounding tissue in the experiments that was not accounted for in the corpuscle-only simulations.

Additionally, the following text was added to the Results:

Accounting for the surrounding tissue in our computational model would better match the simulated tuning curve for the duck HC to the published experimental results shown in Figure 3.

4) The authors agree that the corpuscle may be located at different depths within the skin in different animals. More importantly, the biomechanics of the skin of different animals may be different. How changes in the skin properties affect the optimal frequency of the PC could be added to the text to support the claim.

We thank the reviewer for suggesting this and have added the following text to the Discussion to further emphasize the point that skin biomechanics would affect vibration transmission:

Previous work using this model has shown that the distance between the stimulating probe and a corpuscle embedded in skin affects both the amplitude and phase shift of the vibration transmitted to the receptor's core [17], so any changes in skin depth or mechanics would affect signal propagation through the tissue.

5) The literature for the experimental data quoted in the paper do not have enough details such as the stimulation site, probe, method, frequency etc. I am not sure if such incomplete data can be used for comparison with the simulated data. If used, it is too much of a stretch.

The following text was included in the results to address some of these details and direct readers towards the published manuscript by Gregory, which contains detailed methods:

The neurophysiological data [35] plotted in Figure 3 were obtained from 7 HCs that responded to 1-1075 Hz vibrations induced by a 0.5mm diameter probe on the bill surface. Detailed methods for the experiment can be found in the manuscript by Gregory [35].

6) In spite of so much of variation of PC size, even within a species, for most of the species analyzed in this paper, experimental data from only one published micrograph is used. What is the confidence that this micrograph is a representative data within the species?

We have added the following text to the discussion to address this concern:

We stress that our model for each species is of a single corpuscle, taken from a single published micrograph for that species, and does not involve parameters such as location in the surrounding tissue or interactions with other corpuscles [17]. All of these are significant factors and should be considered in future models, particularly if one were interested in a specific animal and had greater data available. This work illustrates broad trends in corpuscle response but will need further refinement for application to a specific species.

The claim is very interesting, however it needs more supporting evidences.

Appendix C

We thank the reviewer for his/her additional comments. In our response below, the reviewer's comments are given in **boldface**, followed by our response in plain font. Changes to the text are *italicized* below and highlighted in **yellow** in the manuscript.

Associate Editor Comments to Author (Professor Madhusudhan Venkadesan): Comments to the Author:

The responses are satisfactory but the claim in the last sentence of the abstract still concerns me. Given the lack of direct experimental evidence and missing features such as the tissues surrounding the corpuscles, the conclusion of this paper may only be stated as a numerically generated hypothesis. The current concluding sentence of the abstract states the hypothesis as if it's a proven fact. Such a statement runs the risk of future papers claiming that as a result of this work, without appropriate caution. It is important that the claim should be tempered to state that the inter-species conclusion is truly a hypothesis generated by the computational modeling.

The authors thank the Associate Editor for these comments. We have addressed these comments making the following changes to the Abstract, which are italicized below and highlighted in yellow in the revised manuscript:

We observed no correlation between animal size and any measure of corpuscle geometry in our model. Based on the results generated by our computational model, we hypothesize that lamellar corpuscles across different species may utilize different sizes and structures to achieve similar frequency detection bands.